# Bioactive Compounds and Their Influence on Postnatal Neurogenesis

**DOI:** 10.3390/ijms242316614

**Published:** 2023-11-22

**Authors:** Simona Mattova, Patrik Simko, Nicol Urbanska, Terezia Kiskova

**Affiliations:** Department of Animal Physiology, Institute of Biology and Ecology, Faculty of Science, Pavol Jozef Safarik University in Kosice, 041 54 Kosice, Slovakia; simona.mattova@student.upjs.sk (S.M.); patrik.simko1@student.upjs.sk (P.S.); nicol.urbanska@student.upjs.sk (N.U.)

**Keywords:** bioactive compounds, postnatal neurogenesis, neurodegenerative disorders, resveratrol, melatonin, atranorin, EGCG

## Abstract

Since postnatal neurogenesis was revealed to have significant implications for cognition and neurological health, researchers have been increasingly exploring the impact of natural compounds on this process, aiming to uncover strategies for enhancing brain plasticity. This review provides an overview of postnatal neurogenesis, neurogenic zones, and disorders characterized by suppressed neurogenesis and neurogenesis-stimulating bioactive compounds. Examining recent studies, this review underscores the multifaceted effects of natural compounds on postnatal neurogenesis. In essence, understanding the interplay between postnatal neurogenesis and natural compounds could bring novel insights into brain health interventions. Exploiting the therapeutic abilities of these compounds may unlock innovative approaches to enhance cognitive function, mitigate neurodegenerative diseases, and promote overall brain well-being.

## 1. Postnatal Neurogenesis

Neurogenesis, the process of generating functional neurons from precursor cells, was traditionally believed to occur exclusively during embryonic and prenatal stages in mammals [1]. The prevailing consensus for many years was that neurons could not regenerate in the adult brain [2]. However, this viewpoint began to shift with the culmination of several reports on postnatal mitosis in the central nervous system (CNS), notably, in a study by Ezra Allen, an anatomist based in Philadelphia [3]. In the 1960s, Altman’s research provided the initial evidence of neurogenesis in the adult brain. Altman observed the proliferation of glial cells and potential neurogenesis in the brains of mature rats and cats. His studies uncovered labeled neurons in both the dentate gyrus (DG) of the hippocampus and the cortex [4]. In 1969, Altman identified the source of new postnatal granule cell neurons in the olfactory bulb (OB) [5]. In the late 1970s, Kaplan and Hinds conducted autoradiography on tissue slices from normal adult rat brains 30 days after administering tritiated thymidine via intraperitoneal injection. Their research unveiled labeled cells in the OB and the granular layers of the hippocampal DG. With the use of electron microscopy, they were able to confirm that these labeled cells were indeed neurons with observable axons, dendrites, and synapses [6]. Reynolds and Weiss achieved a breakthrough by isolating neural stem cells (NSCs) from the brains of adult mammals [7].

The concept of adult neurogenesis in humans took a significant step forward in 1998 when a study provided compelling evidence of neurogenesis in the human hippocampus. This evidence was obtained by examining postmortem tissue from patients with cancer who were injected with BrdU for therapeutic purposes [8].

In the past decade, there has been a renewed exploration of the mechanisms and extent of neurogenesis in the adult mammalian brain, with studies conducted in mice, rats, monkeys, and humans. It has been established that new nerve cells in the CNS are reliably generated in just two regions: the hippocampus and the OB. In the CNS, new nerve cells primarily take the form of interneurons, including granule cells and periglomerular cells in the OB, as well as granule cells in the hippocampus [9]. However, there is still ongoing speculation about the potential formation of new neurons in other brain regions, including the neocortex, amygdala, hypothalamus, striatum, and substantia nigra [10]. In the mature OB, neurogenic stem cells are located in the anterior subventricular zone (SVZ) of the forebrain, while in the hippocampus, they are found in the subgranular zone (SGZ) within the hippocampal formation [9]. Neurogenesis is most pronounced in the DG of the hippocampus, a brain region crucial for learning and memory. It is fascinating to note that approximately one-third of the neurons in the adult hippocampus are replaced over the course of a person’s lifetime [11].

In the DG, the generation of new cells originates from progenitor or precursor cells located within the DG itself. These cells possess the capacity to proliferate but are more limited in their ability to differentiate compared with stem cells. These progenitor cells exhibit some characteristics like astroglia, a type of glial cell in the brain. Within the SGZ of the DG, cell division occurs, presumably, in an asymmetric manner. This division gives rise to daughter cells, some of which retain the capability to divide further, while others undergo differentiation into either neurons or glial cells. These newly formed neurons then migrate a short distance to the granule cell layer, where they undergo further maturation into functional neurons [12]. These continuously emerging neurons display a remarkable ability to establish new synaptic connections within the existing neural network. This process remodels the circuitry in the hippocampus and represents a unique form of activity-dependent structural plasticity in the mature CNS. Adult hippocampal neurogenesis serves several important functions, including the replacement of redundant neurons, the facilitation of encoding new memories while preventing interference, an enhancement in pattern separation (the ability to distinguish between similar experiences), and the degradation or forgetting of some previously established memories.

The second neurogenic brain region, SVZ, is located within the lateral ventricles. In this area, newly generated neurons predominantly traverse the rostral migratory stream (RMS) to ultimately integrate as interneurons in the OB. The olfactory system, a vital sensory modality, has been remarkably conserved throughout evolution. It serves diverse purposes, such as providing information on food sources, identifying predators and prey, and influencing social and sexual behaviors, including mate selection [13]. Adult neurogenesis in the OB yields various subtypes of interneurons that play a significant role in processing sensory inputs [1]. Neuroblasts exit the SVZ, where they are not guided by radial glia; instead, they migrate tangentially in chains through tubular structures created by specialized astrocytes. After detaching from these chains and migrating radially from the RMS into the OB, most newly generated cells that reach the OB become granule cells and the remainder become periglomerular cells or astrocytes [14]. Granule cells represent the largest proportion of neurons in the OB. Each granule cell establishes connections with several hundred relay neurons, such as mitral and tufted cells, which, in turn, communicate with numerous pyramidal cells in the piriform cortex [15]. The function of adult OB neurogenesis likely includes the maintenance of the fundamental integrity of the OB over time, as well as roles in temporary olfactory memory, olfactory fear conditioning, and long-term associative olfactory memory, including active learning [1].

The process of postnatal neurogenesis, the period before a daughter cell becomes a mature neuron, is an intricate and tightly regulated process that can be divided into six distinct stages: activation, proliferation, differentiation, migration, maturation, and integration. Each of these phases of postnatal neurogenesis is under the influence of internal molecular mechanisms and a variety of factors, with epigenetics also playing a significant role [16]. Interestingly, many of the signaling molecules that induce and specify neural precursors and their progeny during early development in the neural plate and neural tube remain active in adult neurogenic niches. These molecules include Sonic hedgehog (Shh), members of the fibroblast growth factor (FGF) family, TGF-β family members (including bone morphogenetic proteins or BMPs), and retinoic acid. Furthermore, critical transcription factors that are associated with NSCs during the initial development of the CNS, such as Sox2, as well as those that identify transit amplifying cells and newly generated neurons in the embryonic brain, such as the basic helix–loop–helix (bHLH) neurogenic genes, continue to be expressed in adult NSCs and transit amplifying cells [9].

In the SGZ, multipotent NSCs originate from radial glial cells, also known as radial astrocytes. NSCs first give rise to intermediate progenitor cells (type 2 cells), which then generate neuroblasts [17]. The neuroblasts generate immature neurons. These immature neurons subsequently migrate into the inner granule cell layer within the hippocampus, where they differentiate into mature dentate granule cells. Radial astrocytes also generate astrocytes. However, whether they can produce oligodendrocytes in vivo remains uncertain [17]. One critical aspect of this process is the controlled cell death that eliminates most of these cells. Research suggests that there are two crucial periods in the lifespan of these cells during which their survival is particularly significant. The main critical period occurs during the transition from intermediate progenitor cells to neuroblasts when cells exit the cell cycle and most of them die. The second critical period exists at the immature neuron stage. Only neurons that are effectively integrated into the existing neural circuitry are likely to survive beyond this stage [18]. The granule cells in the DG utilize glutaminergic signaling, whereas the granule cells in the OB utilize GABAergic signaling [19].

In addition to cellular factors, environmental factors and pathological conditions are also significantly involved in the regulation of postnatal neurogenesis [1]. The rate of adult neurogenesis, along with the integration of newly generated neurons, is subject to a multifaceted regulation, both positive and negative, with profound implications for brain function [20]. External stimuli, such as physical exercise and cognitive stimulation, can regulate the processes of adult-born neuron production, migration, maturation, and survival [17]. Moreover, adopting a diet rich in nutritious foods and practicing mindfulness-based meditation have been correlated with an expansion in hippocampal volume, which is linked to elevated levels of adult hippocampal neurogenesis. Conversely, the aging process, chronic stress, as well as anxiety and depression, exert detrimental effects on neurogenesis [21]. The aim of the present review is to describe postnatal neurogenesis with its specific regions and to discover the potential of natural compounds during postnatal neurogenesis in physiological or pathophysiological conditions.

## 2. Neurogenic Zones

### 2.1. Hippocampus

The hippocampus is a structure located within the medial temporal lobe of the brain. It assumes a pivotal role in a range of cognitive functions, notably, encompassing learning, memory, and spatial navigation [22]. This bilateral structure consists of two distinct formations, one residing within each cerebral hemisphere. It is primarily composed of the DG, Cornu Ammonis regions (CA1, CA2, CA3), and the subiculum. The intricate interconnections among these regions are critical for the intricate processing and integration of information [23].

### 2.2. Dentate Gyrus

The DG, situated in the posterior region of the hippocampus, serves as the principal input subregion. It receives incoming signals from the entorhinal cortex via the perforant pathway [12]. One of its critical roles is in pattern separation, a process that enables the formation of distinct memory representations by reducing interference between similar experiences [24]. The DG is a specialized subregion within the hippocampal formation. It exhibits unique anatomical and physiological characteristics, which contribute to its specific functions in cognition and memory processing [12]. Several research studies have shown an association between DG alterations and various neurological disorders, including Alzheimer’s disease, epilepsy, and depression [25].

This structure comprises two layers of cells, each with its own distinct functions and structural characteristics. The granule cell layer, predominantly composed of densely packed granule cells, plays a central role. These granule cells receive inputs from the entorhinal cortex via the perforant pathway and project to the CA3 region of the hippocampus. The remarkable properties of granule cells, characterized by their heightened excitability and sparsely distributed activation pattern, significantly contribute to pattern separation and the formation of discrete memory representations [26]. The molecular layer of the DG contains dendrites of granule cells, accompanied by several types of interneurons. Interneurons, including basket cells and hilar mossy cells, assume vital roles in governing the excitability and synchronization of granule cells. Their function lies in exerting control over the DG network by providing inhibition, thereby influencing the overall output of the hippocampus [27].

The DG plays a key role in pattern separation, enabling the distinction of similar input patterns and reducing interference between memories. Both animal models and human neuroimaging studies have shown that the DG contributes to the creation of unique memory representations by generating sparse and orthogonal neural codes [24]. The DG demonstrates remarkable neurogenic abilities that persist into adulthood. NSCs located within the SGZ continually give rise to new neurons, seamlessly integrating them into the existing neural circuitry. This process of neurogenesis in the DG plays a vital role in processes related to learning and memory, as well as in the regulation of mood and emotional responses [28]. Dysfunction of the DG plays an important role in various pathological conditions. In Alzheimer’s disease (AD), the DG emerges as one of the earliest and most profoundly impacted regions. Within this region, the accumulation of neurofibrillary tangles and amyloid-beta plaques leads to synaptic dysfunction, neuroinflammation, and a subsequent decline in cognitive function [29]. The impairment of neurogenesis within the DG may contribute to the memory deficits observed in AD [30]. Numerous studies have reported structural changes within the DG in individuals suffering from depression.

### 2.3. Cornu Ammonis (CA) Regions

The CA1 region of the hippocampus is a crucial component of the brain, responsible for a range of cognitive functions including learning, memory formation, and spatial navigation. The CA1 region possesses a unique anatomical structure within the hippocampal formation [31]. It comprises distinct layers and is composed of principal neurons, specifically known as pyramidal cells, as well as interneurons. The CA1 region is particularly notable for its complex synaptic connectivity and capacity for plasticity. Processes like long-term potentiation (LTP) and long-term depression (LTD) are essential for memory formation and synaptic plasticity within CA1 [32]. Molecular mechanisms, including calcium-dependent processes and signaling pathways, play a significant role in facilitating these synaptic changes [33]. The CA1 region holds a critical role in a variety of cognitive processes, encompassing spatial navigation, contextual memory, and associative learning [34]. Its interactions with other hippocampal subregions, such as CA3 and the DG, contribute to memory recall and the integration of information [35]. The CA1 region has been implicated in the pathophysiology of several neurological disorders. Conditions like depression and anxiety disorders have been associated with aberrations in CA1 function. Investigating these alterations in CA1 function in the context of these disorders may offer valuable insights into potential therapeutic strategies [36].

The CA2 region of the hippocampus is a distinct subregion that has garnered increasing attention in recent years. Positioned between CA3 and CA1, the CA2 region boasts unique anatomical features within the hippocampal formation. Notably, it stands out with its special cellular properties that set it apart from its neighboring regions [37]. Pyramidal neurons are the primary cell type in CA2. They exhibit notable differences in morphology and connectivity patterns when compared with CA1 and CA3 [38]. In terms of synaptic connectivity, CA2 is both complex and distinctive, sharing similarities with other hippocampal subregions while also displaying differences [39]. Recent research has shed light on the role of neuromodulatory systems, such as oxytocin and vasopressin receptors, in governing synaptic plasticity within CA2 [40]. Additionally, CA2 has unique electrophysiological characteristics, including a low threshold for epileptiform activity [37]. Recent evidence indicates that CA2 contributes significantly to various cognitive processes, including pattern separation, social memory, and social behaviors. Its connections with other brain regions, such as the medial prefrontal cortex and amygdala, are critical for the encoding and retrieval of social information [41]. Importantly, dysfunction in CA2 has been implicated in social memory deficits observed in neurodevelopmental disorders and neurodegenerative diseases [40].

The CA3 region of the hippocampus plays a specific role in memory processes, susceptibility to seizures, and neurodegeneration. The CA3 region, positioned between CA2 and the DG, displays notable structural distinctions within the hippocampus. Internal connectivity in the CA3 subfield is richer than in other hippocampal regions. The recurrent axon collaterals originating from CA3 pyramidal cells exhibit a wide-reaching branching pattern, establishing excitatory connections with both neighboring excitatory and inhibitory neurons. This specific neural circuit plays a significant role in the encoding of spatial representations and the formation of episodic memories [42].

### 2.4. Subiculum

The subiculum is a critical region within the hippocampus holding a central role in both spatial navigation and memory consolidation. Functioning as a critical output structure of the hippocampus, it serves as a crucial intermediary connecting the hippocampus to various cortical and subcortical regions [43,44]. The subiculum acts as a bridge, facilitating communication between the hippocampus proper, the entorhinal cortex, multiple cortical areas, and a variety of subcortical structures. Despite its significance, this specific brain structure has been relatively underexplored in research. The subiculum showcases distinct electrophysiological and functional properties that distinguish it from its input areas. Its extensive connections with numerous cortical and subcortical regions grant it the ability to exert influence over a wide range of brain regions, even those that may initially appear unrelated [43].

### 2.5. Subventricular Zone

SVZ is situated along the lateral walls of the lateral ventricles within the cerebral hemispheres [45]. The SVZ is the major source of NSCs in the adult brain of mammals, including humans [46]. In both the hippocampus and the SVZ/OB, the neurogenic process can be subdivided into well-defined stages, from cell proliferation to neuronal differentiation, maturation, and ultimately functional integration, including synaptic connections. Likewise, in the SVZ/OB, adult neurogenesis is thought to contribute to optimal olfactory circuit formation [10]. In adult mammals, newly generated neurons stemming from stem cells in the SVZ embark on a journey through the rostral migratory stream, eventually reaching their destination in the OBs [45]. New OB neurons are thought to contribute to fine odor discrimination and odor–reward association [47].

In SVZ, NSCs are located adjacent to the ependymal cells lining the lateral ventricle. These NSCs, known as type B cells, resemble astrocytes and remain in a quiescent state. They express specific markers such as glial fibrillary acidic protein (GFAP) and CD133. When activated, these type B NSCs divide to produce transit amplifying cells (TACs), also referred to as type C cells. TACs have a high proliferation rate and serve to expand the pool of progenitor cells but have a limited capacity for self-renewal. TACs eventually give rise to neuroblasts, which form chains and migrate out of the V-SVZ into the rostral migratory stream until they ultimately reach the OB [48]. Neuroblasts divide one to two times and migrate through the rostral migratory stream (RMS) toward the OB. Upon reaching the OB, the neuroblasts engage in a radial migration process, guided by various factors like tenascin-2 and prokineticin-2. These factors induce detachment from the RMS chains, allowing the neuroblasts to integrate into the granule cell layer of the OB. Within the OB, they are believed to play a role in plasticity and contribute to OB-dependent learning processes [10].

Once in the OB, neuroblasts originating from the SVZ undergo a crucial transformation. They integrate into the intricate circuitry of the OB and undergo differentiation into interneurons, contributing to the neural network of the OB. The regulation of SVZ-derived cell migration in the adult brain is a complex process involving several mechanisms. This regulation encompasses dynamic cell-to-cell communication, interactions with the extracellular matrix, and the influence of both chemorepellent and chemoattractant signals. Under certain pathological conditions, these processes can be modified to redirect the migration of SVZ-derived cells and provide support in damaged areas [46].

The functional significance of neurogenesis in the SVZ has been somewhat less extensively studied in comparison with hippocampal neurogenesis. Nevertheless, it is noteworthy that SVZ neurogenesis continues to occur throughout adulthood in the mammalian brain and plays a significant role in the development of optimal olfactory circuitry [10]. Additionally, there is evidence to suggest that steroid hormones, specifically estrogen, may exert an influence on SVZ/OB neurogenesis, potentially contributing to sexual function [49]. In particular, these steroid hormones, like estrogen, may be involved in the survival of newly generated OB neurons, which could have implications for the regulation of sexual behavior [10].

### 2.6. Connectivity and Circuitry

The hippocampus is highly regarded for its intricate connectivity with various other regions of the brain. A prominent feature of this connectivity is the entorhinal cortex, which serves as the primary source of input to the hippocampus. Conversely, the hippocampus sends its output to several areas, including the prefrontal cortex, amygdala, and other regions [50]. The trisynaptic circuit, consisting of the perforant pathway, mossy fibers, and Schaffer collaterals, forms the core circuitry within the hippocampus [26].

Understanding the anatomy of the hippocampus is essential for unraveling its functional significance. The hippocampus comprises several key components, including the DG, the CA regions (CA1, CA2, CA3), and the subiculum. These elements collectively constitute the complex architecture of the hippocampus. The connections and circuitry within the hippocampus are of paramount importance, as they underpin its pivotal role in processes such as learning, memory formation, and spatial navigation. The intricate interplay of these hippocampal regions and their connections allows for the integration and processing of information critical to these cognitive functions.

### 2.7. Rostro-Caudal Migration Pathway

During CNS development, neurons embark on a remarkable journey, navigating from their initial places of origin to their ultimate destinations. The process of neuronal migration is of utmost importance as it is instrumental in establishing the intricate circuitry that characterizes the brain. One of the prominent migration pathways that has garnered considerable attention is the rostro-caudal migration pathway. This pathway plays a pivotal role in the precise positioning of specific neuronal populations within the brain.

The rostro-caudal migration pathway involves the migration of neurons from the rostral (anterior) regions to the caudal (posterior) regions of the developing brain. This migration process contributes significantly to the organization and functional specialization of various brain regions, ultimately shaping the complex architecture of the CNS [51].

Neuronal migration is a fundamental process in the intricate development of the brain, playing a crucial role in ensuring that neurons find their appropriate positions within the developing brain. While the radial and tangential migration pathways have long been recognized and well-studied, the rostro-caudal migration pathway has recently gained prominence as a significant mechanism in shaping the architecture of the brain.

The rostro-caudal migration pathway is characterized by the movement of neurons along the anterior–posterior axis, contributing substantially to the regionalization and specialization of various brain areas. To fully grasp the complexity of brain wiring, it is essential to comprehend the cellular and molecular mechanisms that govern rostro-caudal migration. This knowledge is vital for unraveling the intricate processes that underlie the development and organization of the brain [52].

Rostro-caudal migration is a highly intricate process involving a series of complex cellular processes that direct the movement of neurons along the anterior–posterior axis of the developing brain. This migration relies on the precise coordination of multiple cellular processes, including the rearrangement of the cytoskeleton, cell adhesion, and signaling pathways associated with migration. These glial cells extend their processes from the ventricular zone to the pial surface of the brain, effectively creating a physical scaffold for neuronal migration. Neurons then utilize these glial fibers as their migration routes, relying on adhesion molecules and the dynamic behavior of the cytoskeleton to navigate through the intricate terrain of the developing brain [53].

The intricate process of rostro-caudal neuronal migration is subject to the regulation of various molecular cues and signaling pathways that ensure the precise positioning of neurons within the developing brain. One of the notable molecular players in this process is Sonic hedgehog (Shh), which exhibits a graded expression pattern along the rostro-caudal axis [54]. Shh acts as a chemoattractant for migrating neurons, guiding them toward regions with higher concentrations of this ligand.

In addition to Shh, guidance cues such as Netrin-1 and Slit proteins, along with their corresponding receptors, DCC and Robo, respectively, contribute significantly to the accurate navigation of migrating neurons along the rostro-caudal axis [55]. Furthermore, intracellular signaling pathways, including the Reelin-Dab1 and Wnt-β-catenin pathways, play essential roles in coordinating the various processes involved in neuronal migration [56].

## 3. Disorders Associated with Suppressed Neurogenesis

Changes in adult neurogenesis and a reduction in hippocampal size have been observed in numerous psychiatric disorders and several neurodegenerative diseases [57]. For instance, neurodegenerative conditions associated with aging, such as Alzheimer’s disease and Parkinson’s disease, can impair adult hippocampal neurogenesis [58]. Moreover, multiple sclerosis can also have a detrimental impact on hippocampal function [59]. The intricate connections between the hippocampus and other structures in the limbic system mean that mental states such as anxiety and depression can affect adult hippocampal function and neurogenesis. These findings highlight the critical role of the hippocampus in various neurological and psychiatric conditions [60].

### 3.1. Alzheimer’s Disease

Alzheimer’s disease (AD) stands as the most prevalent form of neurodegenerative dementia globally and is characterized by early impairment of recent memory. As the disease advances in severity, individuals with AD experience a wide array of symptoms, including impaired language, orientation, and impaired executive functions, leading to a decline in their ability to care for themselves [61]. The entorhinal cortex and the hippocampus play a key role in AD etiology. The entorhinal cortex is among the first brain regions affected in AD, followed by subsequent involvement of the hippocampus and cerebral cortex [62]. Specifically, AD is linked to the formation of extracellular insoluble β-amyloid (Aβ) deposits, known as Aβ plaques, and the intraneuronal accumulation of abnormally phosphorylated tau protein, referred to as tau neurofibrillary tangles. These plaques and tangles are closely associated with neuronal cell death and brain atrophy [63]. The presence of plaques and tangles within the hippocampus is strongly correlated with cognitive decline [62].

### 3.2. Parkinson’s Disease

Parkinson’s disease (PD) is the most common movement disorder with an increasing prevalence because of an aging population. Its motor manifestations encompass bradykinesia, rigidity, resting tremor, and postural instability, primarily attributed to the deterioration of dopaminergic neurons in the substantia nigra pars compacta. Meanwhile, nonmotor symptoms such as depression, anxiety, and cognitive and olfactory impairments, as well as autonomic dysfunction, are likely linked to disruptions in other neurotransmitter systems [64]. In PD, the gradual demise of dopaminergic neurons within the substantia nigra pars compacta leads to progressive degeneration and dysfunction of the nigrostriatal pathway. Dopamine levels in the striatum are significantly reduced, and the loss of neuronal communication between the two regions subsequently impairs the brain’s ability to produce smooth and purposeful movements. Clinically, this manifests as the progressively severe motor symptoms characteristic of PD [65]. One critical role of α-synuclein lies in the regulation of presynaptic transmission. Elevated α-synuclein expression has been associated with diminished neurogenesis and impaired morphological maturation of adult-born DG cells. Consequently, maintaining appropriate levels of α-synuclein is vital for the proper control of adult hippocampal neurogenesis. Genetic mutations observed in PD patients may thus influence adult hippocampal neurogenesis, contributing to at least a portion of PD’s pathology [58]. Hippocampal dysfunction is a common occurrence in PD patients and is likely a contributing factor to depression and cognitive impairments [66].

### 3.3. Multiple Sclerosis

Multiple sclerosis (MS) is a chronic inflammatory disorder of the CNS characterized, particularly in its early stages, by a significant influx of immune cells, notably, helper CD4 + T cells, and extensive activation of microglial cells. This immune response results in the demyelination of axons, neuroaxonal degeneration, and brain atrophy, which becomes increasingly prominent as the disease advances [67]. Neuroinflammation, instigated by the activation of microglia, primarily targets myelin, which is a key neuropathological feature of MS. However, this neuroinflammatory cascade also inflicts damage on neurons and synapses, leading to demyelination, atrophy, and degeneration of gray matter. It is worth noting that a significant proportion of individuals with MS experience clinically relevant cognitive impairments throughout the course of the disease, including difficulties with learning and memory. This suggests the involvement of the hippocampal memory system in the disease’s pathogenesis [59]. Demyelination has been noted in gray matter regions, including the hippocampus, deep gray matter, cerebellar cortex, and spinal gray matter [68]. In addition to a reduction in hippocampal volume, there is evidence of compromised connectivity involving the hippocampus. This is observed with resting-state functional connectivity measurements between the hippocampus and regions like the anterior cingulate gyrus, thalamus, and prefrontal cortex. These findings indicate impairments in large-scale cognitive networks [59].

### 3.4. Dementia

Dementia refers to a condition characterized by a significant decline in cognitive function from one’s previous level, leading to disruptions in occupational, domestic, or social activities [69]. Dementia with Lewy bodies (LB) is characterized clinically by a predominant dementia syndrome preceding motor symptoms and pathologically by the neocortical accentuation of LB pathology. In individuals with PD, the extent of LB pathology within the hippocampus has been found to correlate with the severity of dementia symptoms. Magnetic resonance imaging studies of patients with Parkinson’s disease dementia have revealed significant atrophy of the hippocampus when compared with those with Parkinson’s disease who do not exhibit dementia symptoms [66].

### 3.5. Depression

Depression is a mental disorder that exhibits both a high incidence and substantial disability rates. According to the Global Burden of Disease Study, depression is now the second most disabling condition worldwide. The heterogeneity observed in depression underscores the intricate nature of research into this condition. One of the primary risk factors for depression is stress, which can trigger neuroinflammatory responses leading to anxiety and behaviors reminiscent of depression [70]. Neuroinflammation has a profound impact on cognitive function and contributes to neuronal loss, particularly affecting the process of neurogenesis in the hippocampus. This pro-inflammatory state in the adult hippocampus is closely linked to increased expression of glucocorticoids. The equilibrium between pro-inflammatory and anti-inflammatory signaling pathways plays a pivotal role in regulating adult hippocampal neurogenesis [45].

Depression is associated with alterations in monoaminergic neurotransmitter systems in various brain regions, as well as the suppression of hippocampal neurogenesis. These changes can lead to disruptions in the activity of brain regions related to cognition and emotion. Moreover, depression is often associated with dysfunction in the hypothalamic–pituitary–adrenal (HPA) axis, which can exacerbate the effects of stress, including its impact on serotonin (5-HT) activity [71]. Many antidepressants have been shown to stimulate adult hippocampal neurogenesis [45].

Neurogenesis plays a critical role in the potential treatment of mood disorders, primarily because of the notable changes that occur in the hippocampus of individuals with depression and other affective disorders. Cognitive deficits, mood instability, and reduced hippocampal volume are all correlated with mental disorders that show reduced neurogenesis, such as major depression, post-traumatic stress disorder, schizophrenia, and Alzheimer’s disease [72].

## 4. Neurogenesis-Stimulating Bioactive Compounds

Nature serves as a rich source of fascinating compounds, with some approved drugs like paclitaxel and morphine originating from plants. Notably, a substantial portion of drugs introduced over the past two decades are either based on natural products or derived from them synthetically [73].

### 4.1. Plant Extracts

Throughout history and even today, nature continues to provide valuable foods and ingredients that promote human health. In the contemporary food industry, plant extracts are gaining significance as essential additives, thanks to their rich content of bioactive compounds [74]. Some of them, such as saffron, lavender, basilicum, and rosa, are discussed in this review.

#### 4.1.1. Saffron

There is increasing evidence that the consumption of saffron, a spice derived from the flower of the *Crocus savitus* L., offers numerous therapeutic benefits, including protection of the CNS [75]. Saffron has historical applications in treating depression and various inflammatory conditions in ancient China, owing to its antioxidant, anti-inflammatory, and antidepressant properties [76]. Saffron is well-known for its antidepressant effects [77], which have been extensively evaluated in various clinical trials [78]. It has beneficial effects on neuropsychiatric, neurodegenerative, and other brain-related disorders, as previously reviewed [76]. In vitro, saffron (or its parts such as crocetin or crocin), has demonstrated the ability to enhance the proliferation of NSCs [79]. In vivo, saffron alone or together with exercise significantly increases the levels of brain-derived neurotrophic factor (BDNF) and serotonin in the hippocampus compared with control Wistar rats. This elevation is associated with improved short-term memory and increased neurogenesis [80].

#### 4.1.2. *Lavandula* sp.

*Lavandula* sp. has been used for centuries for its sedative and relaxation-inducing properties, primarily as an aromatherapy agent [81]. Lavender oil has exhibited antidepressant effects [82] and caused a significant reduction in parameters such as blood pressure, heart rate, and skin temperature, indicative of a decrease in autonomic arousal [81]. It also demonstrated anxiolytic potential by altering behavior in an open field test, comparable to the effects of chlordiazepoxide. Additionally, lavender oil was found to reduce c-fos expression in the paraventricular nucleus of the hypothalamus and dorsomedial hypothalamic nucleus [83]. Lavender is believed to inhibit sympathetic nerve activity and lipolysis through the activation of H3-receptors [84]. However, despite numerous clues and indications, there have been no recent studies focusing on the impact of lavender on postnatal neurogenesis for an extended period. In 2019, Sánchez-Vidaña et al. presented pioneering research, demonstrating that a 14-day-long lavender aromatherapy regimen reversed the depression-like and anxiety-like behavior induced by high corticosterone levels in male Sprague Dawley rats. Furthermore, this treatment was associated with improvements in neurogenesis and dendritic complexity within the hippocampus and SVZ of these animals [85].

#### 4.1.3. Ocimum Basilicum

*Ocimum basilicum* has been reported to possess antidepressant and anxiolytic properties [86]. It was shown that during chronic unpredictable mild stress (CUMS)-induced depression, *Ocimum basilicum* effectively mitigated several adverse effects. Specifically, it ameliorated CUMS-induced hippocampal neuron atrophy and apoptosis while also promoting the proliferation of astrocytes and new nerve cells. Furthermore, *Ocimum basilicum* significantly lowered corticosterone levels and upregulated the expression of both proteins and genes related to brain-derived neurotrophic factor (BDNF) and glucocorticoid receptors [87]. In male Swiss albino mice exposed to CUMS, *Ocimum basilicum* not only alleviated behavioral and biochemical alterations but also reduced neuronal atrophy in the hippocampal CA3 and DG regions. Additionally, it stimulated neurogenesis in a pattern similar to that of fluoxetine, a commonly prescribed antidepressant medication [88].

#### 4.1.4. Rosa Damascena

*Rosa damascena* possesses various pharmacological effects, including anti-HIV, antibacterial, antioxidant, antitussive, hypnotic, and antidiabetic effects, as reviewed in [89]. Moreover, the extract from *Rosa damascena* showcases distinctive characteristics when it comes to its impact on the CNS. It has been identified to possess anticonvulsant and neuroprotective effects, effectively preventing the formation of dark neurons [90], which are commonly associated with epileptic seizures or brain ischemia [91]. Additionally, *Rosa damascena* extract significantly prolongs the latency of seizure attacks and diminishes the frequency and amplitude of epileptiform burst discharges induced by pentylenetetrazole injection [90]. The hydro-alcoholic extract of *Rosa damascena* prevented anxiety behavior as tested in the elevated plus maze. It augments the activity of antioxidant enzymes such as superoxide dismutase and glutathione peroxidase while reducing the levels of serum corticosterone in Wistar male rats treated with a high-fat diet [92]. *Rosa damascena* extract in doses of 300, 600, and 1200 mg/kg improved spatial learning and memory in a dose-dependent manner in a model of Alzheimer’s disease. Concurrently, the extract promotes the proliferation of NSCs and neural progenitor cells (NPCs) in the hippocampus across all extract-treated groups. Additionally, *Rosa damascena* reverses the age-related decline in hippocampal volume and restores the absolute volumes of the DG and CA1 layer [93].

### 4.2. Bioactive Substances

It was determined that the activity of plant extracts is linked to a specific organic compound, possessing its distinct identity and the potential for purification. This discovery marked the commencement of natural product research, and to this day, plant secondary metabolites have been instrumental. This is evident in the fact that over 30% of medicinal products are directly or indirectly derived from natural products [94]. Only from the group of flavonoids, more than 4000 different compounds have been identified from plant origins to date [94]. In this review, only a selected few are described.

#### 4.2.1. Melatonin

Numerous natural compounds have shown potential as bioactive substances that can stimulate neurogenesis. One such compound is melatonin, a well-known hormone produced by the pineal gland. Melatonin can also be found in external sources like almonds, white radishes, rice, coffee, cherries, strawberries, and more [95]. Melatonin is known for its antioxidative [96] and anticancer properties [97]. Traditionally, melatonin is associated with the regulation of circadian rhythms and the sleep–wake cycle. However, it has been discovered that melatonin also plays a crucial role in regulating neurogenesis, making it a possible treatment option for neurogenesis-related disorders [98]. In vitro, melatonin in the rising concentrations of 0.05–10 μM was able to promote the viability, proliferation, and neuronal differentiation of mice embryonic cortical NSCs [99]. This effect is likely mediated through an increase in the acetylation of histone H3 lysine 14 (H3K14). Furthermore, this enhanced H3K14 acetylation alters the chromatin structure of the promoters of basic helix–loop–helix factors Neurogenin1 and NeuroD1, thereby activating their transcription. Melatonin achieves this by activating the histone acetyltransferase activity of CREB-binding protein (CBP)/p300 via the ERK signaling pathways [100]. Furthermore, melatonin has been found to promote the proliferation of NSCs from the adult mice spinal cord and this effect is likely mediated through the PI3K/AKT signaling pathway [101]. It seems that the activity of melatonin is enabled via melatonin receptors MT1 and MT2 [102], which are distributed throughout the hippocampus, with a particular concentration in the hippocampal neurogenic microenvironment or niche. This suggests that these receptors play a role in melatonin’s beneficial effects on hippocampal neurogenesis and behavior [103]. Additionally, melatonin receptors have been identified in various other brain regions, including oligodendrocytes in the corpus callosum, the SVZ where neural stem/progenitor cells (NSPCs) are found, and the choroid plexus, which acts as a barrier between the blood and cerebrospinal fluid [104]. In vivo studies have shown that one-month-long delivery of melatonin in drinking water (4 mg/kg/day) after 24 h stroke onset influences neuronal survival and stimulates neurogenesis in C57Bl6/j mice, leading to improvements in motor and coordination deficits (assessed using the RotaRod test) as well as behavioral adjustments (assessed in the open field test) [105]. Melatonin, administered at the same dosage and method, has demonstrated protective effects in prenatally irradiated rats (exposed to 1 Gy of radiation). In male Wistar rats, there was a significant increase in the number of BrdU-positive cells in the hilus region compared with the irradiated group that did not receive melatonin treatment. Additionally, there was a marked increase in the number of mature NeuN-positive neurons in the hilus and granular cell layer of the DG, as well as in the CA1 region of adult irradiated rats, when melatonin was administered. Furthermore, melatonin significantly improved spatial memory that had been impaired with radiation exposure, as assessed in the Morris water maze test. Notably, a significant correlation was found between the number of proliferative cells and cognitive performance in these rats [106]. In 1-month-old male pinealectomized SD rats, melatonin administered at a concentration of 6 mg/L in drinking water led to a significant increase in the number of doublecortin (DCX) neurons in the DG. This effect was most pronounced after 6 months post-surgery [107]. During depressive-like states in laboratory animals, the administration of exogenous melatonin at a dosage of 8 mg/kg increased the survival of neuronal progenitors and immature neurons in the DG of adult C57BL/6 mice. This was accompanied by antidepressant-like effects observed in the Porsolt forced swim test [108]. Furthermore, in male BALB/c mice subjected to chronic mild stress to induce depression, melatonin (at a dosage of 2.5 mg/kg) and citalopram (an antidepressant drug belonging to the selective serotonin reuptake inhibitors class, administered at a dosage of 5 mg/kg) exhibited similar antidepressant-like activities. These effects were associated with microglial remodeling, CX3C chemokine activity, and enhanced cell proliferation and survival of DCX-positive cells in the DG [109]. Melatonin has shown beneficial effects in Alzheimer’s disease, including an increase in β-catenin protein expression and activation in human neuroblastoma cell cultures [110], as well as the stimulation of neurogenesis [111]. However, when it comes to multiple sclerosis (MS), recent data have presented conflicting results regarding the influence of melatonin. In the MS context, exposure to constant light has been found to promote the proliferation of NSCs and their differentiation into oligodendrocyte precursor cells (OPCs), as well as the maturation of OPCs into oligodendrocytes and their recruitment to sites of demyelination. Additionally, constant light increases the number of patrolling monocytes and enhances their phagocytic activity. In contrast, constant darkness and exogenous melatonin have been shown to exacerbate these events and amplify monocyte infiltration in the context of MS. Therefore, melatonin should not be viewed as a one-size-fits-all remedy. It is crucial to monitor melatonin and cortisol levels in each MS patient, taking into consideration their diet and lifestyle, to avoid melatonin overdose. Individualized approaches are essential to ensure safe and effective treatment. This emphasizes the importance of tailoring treatments to the specific needs and conditions of patients, especially when dealing with complex and multifaceted diseases like multiple sclerosis [104].

#### 4.2.2. Resveratrol

Another well-known natural compound is resveratrol, present in some plants such as red grapes or red wine, which is recognized for its protective properties against vascular diseases, neurodegenerative conditions, atherosclerosis, oxidative stress, and certain types of tumors, primarily due to its antioxidant effects [112]. Resveratrol acts as a phytoestrogen, and its action is dependent on estrogen receptors (ER) [113]. Membrane-associated ERs have been identified in various brain regions involved in learning and memory, including the prefrontal cortex, dorsal striatum, nucleus accumbens, and hippocampus [114]. In the context of Alzheimer’s disease, resveratrol has demonstrated the ability to prevent memory loss, as measured with the object recognition test in AβPP/PS1 mice. Additionally, it reduced the amyloid burden in these mice. These effects were mediated by the increased activation of the Sirtuin and AMP-activated protein kinase (AMPK) pathway [115]. Resveratrol has also been shown to induce neurogenesis and mitochondrial biogenesis through the enhancement of AMPK, independent of SIRT activation [116]. Furthermore, in an in vitro study using rat hippocampal H19-7 neuronal cells, resveratrol exhibited a neuroprotective effect. A two-hour pretreatment with resveratrol at a concentration of 75 µM reduced oxidative damage caused by Aβ and prevented the decrease in crucial proteins associated with synaptic maturity and plasticity [117]. Neurogenesis and synaptic plasticity were normalized after resveratrol administration in a model of streptozotocin-induced diabetes in C57Bl/6 mice. This effect was mediated through the involvement of SIRT1 and AMPK. Additionally, a genome-wide gene expression analysis revealed that resveratrol normalized the hippocampal expression of genes associated with neurogenesis and synaptic plasticity, including Hdac4, Hat1, Wnt7a, and ApoE [118]. During aging, resveratrol improved memory, learning, and mood activities in male F344 rats for 25 months. Resveratrol-treated animals also displayed positive effects on neurogenesis and microvasculature in the hippocampus. It also reduced astrocyte hypertrophy and microglial activation in this brain region [119]. On the other hand, there is also evidence that resveratrol in the dose of 20 and 50 μM may inhibit the proliferation of in vitro NSCs, potentially inducing necrotic cell death. In an in vivo study using 57BL/6 mice, resveratrol at doses of 1 and 10 mg/kg decreased the proliferation of newly generated cells in the hippocampal DG in a dose-dependent manner. Importantly, this decrease in cell proliferation occurred without activating glial cells or causing neuronal damage [120].

#### 4.2.3. Epigallocatechin-3-Gallate (EGCG)

Another natural compound found to have neurogenic potential is epigallocatechin-3-gallate (EGCG), the primary polyphenol found in green tea [121]. EGCG has been shown to have neuroprotective properties [122]. It prevents against amyloid aggregation [121]. In a study involving mice with lipopolysaccharide-induced neuroinflammation, EGCG at a concentration of 0.5 mg/kg was found to restore neurogenesis in the DG. Additionally, EGCG treatment attenuated the production of pro-inflammatory cytokines induced by lipopolysaccharides by modulating the TLR4-NF-κB pathway [123]. In another study, a two-week treatment with EGCG at varying doses (0, 0.625, 1.25, 2.5, 5, and 10 mg/kg) dose-dependently increased the survival of NSCs. This treatment also led to an increase in the population of DCX-expressing cells, indicating enhanced neurogenesis in the adult hippocampus. Furthermore, EGCG treatment was associated with increased levels of phospho-Akt in the hippocampus [124].

#### 4.2.4. Quercetin

Flavonoids and their derivatives, including 7,8-dihydroxyflavone derivatives, have been proven for their potential neuroprotective possibilities [112,113,114]. Quercetin, a naturally occurring flavonoid classified within the flavonol subfamily, is well-known for its anti-inflammatory properties [115]. It has been found to protect neurons from oxidative damage and reduce lipid peroxidation. Additionally, quercetin, when administered at a dose of 10 μg/mL in ethanol solution, inhibits the fibril formation of Aβ proteins associated with Alzheimer’s disease in both male and female Swiss white mice [116]. In a mouse model of Alzheimer’s disease (3xTg-AD mice), quercetin was found to protect the neuronal population in the subiculum, reverse β-amyloidosis in various cerebral regions, and reduce tauopathy (see Figure 1). Furthermore, quercetin treatment led to improved performance in spatial learning and memory tasks and a reduction in anxiolytic behavior [117]. Quercetin has also been shown to increase the proliferation of hippocampal neurons by enhancing the levels of CREB (cAMP response element-binding protein) and BDNF (brain-derived neurotrophic factor) in the brains of rats with induced Alzheimer’s disease [118]. In another study, quercetin at the dose of 40 mg/kg/day increased neurogenesis by promoting the expression of BDNF, NGF, CREB, and EGR-1 genes in the adult rat DG. These gene expression changes were associated with increased maturation of NSCs into neural lineages. Importantly, this enhanced neurogenesis was accompanied by improvements in memory [119].

#### 4.2.5. Atranorin

Atranorin, a compound belonging to the group of lichen secondary metabolites, has garnered interest in recent years due to its intriguing biological activities [125]. Atranorin, along with other lichen secondary metabolites like perlatolic, physodic, and usnic acid, demonstrated neurotrophic activity in a preliminary cell-based screening based on Neuro2A neurite outgrowth. Importantly, atranorin exhibited no cytotoxic effects and was found to modulate the gene expression of brain-derived neurotrophic factor (BDNF) and nerve growth factor (NGF) [126]. Because studying lichen metabolites in vivo is a rare matter, to date, there exist only two studies indicating its action during depressive-like behavior in laboratory animals [125] including rearing frequency and prolonged time spent in open arms in the elevated plus maze test when compared with non-treated depressive Wistar rats. Additionally, in the open field test, the moving speed and trajectory of the treated rats returned to levels similar to those of healthy animals. This was accompanied by an increase in hippocampal neurogenesis in the hilus and SGZ, as well as an increase in the number of mature NeuN neurons in the hilus and CA1 regions of the brain. These effects are believed to be mediated, at least in part, by atranorin’s antioxidant properties [127].

#### 4.2.6. Ginsenoside Rg1

Ginsenoside Rg1 (Rg1) is one of the major bioactive ingredients of *Panax ginseng* [128]. Among others, Rg1 has been shown to have neuroprotective properties [129]. These properties make Rg1 a subject of interest in research exploring potential therapeutic applications in the treatment of neurodegenerative diseases [130]. In a study by Zhu et al. (2014), ginsenoside Rg1 administered at a dose of 20 mg/kg exhibited neuroprotective effects on the hippocampus of three-month-old male Sprague Dawley rats with D-galactose-induced aging. Rg1 mitigated the age-related changes in the hippocampus, including improvements in cognitive capacity, a reduction in senescence-related markers, and an enhancement in hippocampal neurogenesis. Additionally, Rg1 increased hippocampal cell proliferation, enhanced the activity of antioxidant enzymes like glutathione peroxidase and superoxide dismutase, and decreased the levels of proinflammatory cytokines. These findings suggest Rg1′s potential in countering age-related cognitive decline and neuroinflammation [131]. The antidepressant effects of Rg1 were assessed in male C57BL/6J mice. Even a single injection of Rg1 produced a robust antidepressant effect in the forced swim test. Rg1 treatment demonstrated a dose-dependent reduction in the duration of immobility in both the tail suspension and forced swim tests. The antidepressant-like effects of Rg1 were associated with changes in the levels of pERK1/2 and pCREB, indicating that Rg1-induced alterations in the BDNF signaling pathway in the hippocampus may contribute to its behavioral effects. This suggests that Rg1 may have potential as an antidepressant agent [132]. Selected bioactive compounds and molecular pathways leading to postnatal neurogenesis stimulation are listed in Table 1.

## 5. Conclusions

Within the realm of drug discovery, natural compounds assume a central role, serving as a wellspring of chemical diversity that possesses the capacity to ignite the creation of groundbreaking therapeutic agents. These compounds provide a foundation for developing drugs that target a wide range of diseases, including neurological disorders. By harnessing the intricate structural complexities and evolutionary optimizations inherent in these natural molecules, researchers are poised to expedite the advancement of novel treatments, potentially resulting in reduced side effects, and enhanced therapeutic outcomes.

## Figures and Tables

**Figure 1 ijms-24-16614-f001:**
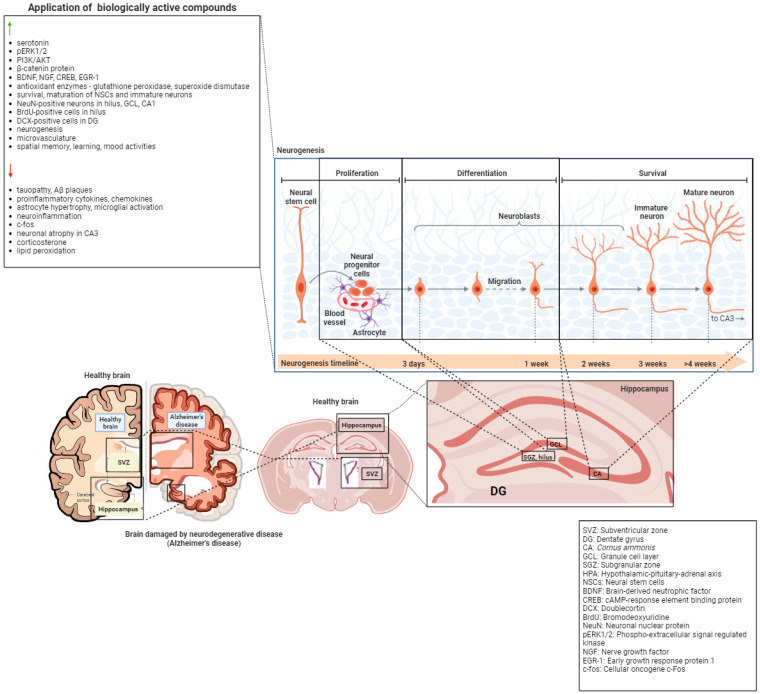
The application of biologically active compounds has an impact on molecular mechanisms involved during postnatal neurogenesis not only in healthy brains but also during various neurological diseases. Created with BioRender.com.

**Table 1 ijms-24-16614-t001:** Selected bioactive compounds and molecular pathways leading to postnatal neurogenesis stimulation.

Compound	Dose	Model	Molecular Activity	Ref.
melatonin	0.05–10 µM	NSCs from mouse embryonic E14 cortex	promotes viability, proliferation, and neuronal differentiation NSCs increase in MBP-positive cells (oligodendrocytes)	[99]
	0.01–10 µM	NSCs from adult mouse spinal cord C57BL/6	increase in neurosphere formation increase in mRNA expression of Sox2 and Hes5 (markers of NSCs) promotes proliferation via melatonin receptors increase in phosphorylation level AKT PI3K/AKT signaling pathway	[101]
	2.5 mg/kg	male BALB/c mice depression model	microglial remodeling CX3C chemokine activity enhancement in cell proliferation survival of DCX+ cells in the DG	[109]
	4 mg/kg	Wistar rats (irradiated rats)	improves spatial memory impaired by irradiation increase in BrdU-positive (proliferative) cells increase in mature NeuN-positive neurons	[106]
	6 mg/L	Sprague Dawley rats pineactomized	increase in DCX neurons (newborn neurons) in DG	[107]
	8 mg/kg	adult C57BL/6 mice depression model	modulates the survival of newborn neurons increase in survival of NSCs and immature neurons in DG increase in BrdU/DCX-labeled cells increase in BrdU/DCX/CR-labeled cells	[108]
resveratrol	75 µM	H19-7 rat cells (Aβ-induced oxidative stress and memory loss)	reduction in oxidative damage improvement in the expression of memory-associated proteins	[117]
	16 mg/kg	AβPP/PS1 mice Alzheimer model	memory loss prevention Sirtuin and AMPK pathway	[115]
	50 mg/kg	C57Bl/6 mice diabetes model	synaptic plasticity neurogenesis stimulation via Hdac4, Hat1, Wnt7a, and ApoE suppression of the Jak-Stat signaling pathway (pro-inflammatory signaling)	[118]
	40 mg/kg	male F344 rats aging model	memory, learning, and mood improvement neurogenesis stimulation astrocyte hypertrophy reduction increase in DCX1 newly born neurons	[119]
EGCG	2.5 mg/kg	NSCs from Balb/C mice (ex vivo)	survival and neuronal differentiation of adult hippocampal precursor cells	[124]
	2.5 mg/kg	male Balb/C mice	increased cell survival increase in DCX+ cells increase in hippocampal levels of phospho-Akt	[124]
	0.5 mg/kg	male C57BL/6 mice (LPS-induced neuroinflammation model)	stimulation of neurogenesis in DG suppression of the activity of microglia and the TLR4-mediated NF-kB pathway (anti-inflammatory activity)	[123]
quercetin	25 mg/kg	3xTg-AD mice Alzheimer model	reduction in taupathy neuronal population protection improvement in spatial learning	[133]
	40 mg/kg	male Wistar rats Alzheimer model	increase in neurogenesis increase in the number of proliferating NSCs/progenitor cells increase in DCX+ cells improvement in the number of BrdU/NeuN+ cells promotes the expression of BDNF, NGF, CREB and EGR-1 genes in adult rat DG	[134]
	100 mg/kg	3xTg-AD mice Alzheimer model	increase in neurogenesis enhances the levels of CREB and BDNF β-amyloidosis reduction and tendency to decrease tauopathy	[135]
atranorin	5 µM	Neuro2A neurite outgrowth	neurotrophic activity increase in BDNF and NGF	[126]
	10 mg/kg	healthy Wistar rats	changed some forms of behavior promotes neurogenesis	[125]
	10 mg/kg	Sprague Dawley rats depression model	changed some forms of behavior promotes neurogenesis	[136]
Rg1	2.5–20 mg/kg	C57BL/6J mice depression model	antidepressant activity changed some forms of behavior pERK1/2 and pCREB pathway	[132]
	20 mg/kg	Sprague Dawley rats aging model	neurogenesis stimulation improvement in cognitive capacity protect NSCs/progenitor cells by increased level of SOX-2 expression reduced astrocytes activation by decrease level of Aeg-1 expression anti-inflammatory activity enhances the activity of antioxidant enzymes GSH-Px and SOD	[131]

Aeg-1, astrocyte elevated gene-1; AKT, protein kinase B; AMP, AMP-activated protein kinase; ApoE, apolipoprotein E; BDNF, brain-derived neurotrophic factor; BrdU, bromodeoxyuridine; CR, calretinin; CREB, cyclic AMP-responsive element-binding protein 1; DCX, doublecortin; DG, DG; EGR-1, early growth response protein 1; GSH-Px, glutathione peroxidase; Hat1, histone acetyltransferase 1; Hdac4, histone deacetylase 4; Hes5, Hes family BHLH transcription factor 5; JAK/STAT, Janus kinase-signal transducer and activator of transcription; MBP, myelin basic protein; NeuN, neuronal nuclei; NGF, nerve growth factor; NSCs, neural stem cells; pERK1/2, phospho extracellular signal-regulated kinase 1/2; PI3K/AKT, phosphatidylinositol 3-kinase (PI3K)/protein kinase B (AKT); Sox2, SRY-Box transcription factor 2; SOD, superoxide dismutase; TLR4-NF-κB, Toll-like receptor 4/nuclear factor kappa B; Wnt7a, Wnt family member 7A.

## Data Availability

Not applicable.

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
