# Peer review of "Bioactive Compounds and Their Influence on Postnatal Neurogenesis"

_ijms, 2023, doi:10.3390/ijms242316614_

Round 1

Reviewer 1 Report

Comments and Suggestions for Authors

Mattova et al. Bioactive compounds and their influence on postnatal neurogenesis

This review covers postnatal neurogenesis, neurogenic zones, disorders with reduced neurogenesis, and bioactive compounds that have the potential to stimulate neurogenesis. Leveraging these compounds could offer novel strategies to improve cognition, address neurodegenerative conditions, and support brain health in general. The focus was on the beneficial effects of bioactive compounds such as Melatonin, resveratrol, EGCG, Quercetin, atranorin, Rg1, Saffron, Lavender, Sweet Basil, and Rosa damascena on brain health. Although the article is well-written and provides an in-depth account of various bioactive compounds, the following are important suggestions to improve the article:

A detailed account of bioactive compounds may be written as subsections 2.14.1, 2.14.2, and so on.

Lines 67 to 117 give an account of neurogenesis in the DG. It would be great to add a brief account of neurogenesis in the olfactory bulbs and its roles (a brief of section 2.5).

It would be beneficial to cite more original research and review articles in reputed journals rather than citing books so that readers could cross-validate the references. It is hard to verify book citations for readers.

Line 89: The process of postnatal neurogenesis, the period before the daughter cell becomes a mature neuron, lasts approximately 6-8 weeks [19]. This statement is not clear as it doesn’t mention if the experiments were done on humans or animals.

Please add all of the following compounds (Melatonin, resveratrol, EGCG, Quercetin, atranorin, Rg1, Saffron, Lavender, Sweet Basil, and Rosa damascena) in the table; some of them are missing.

Some of the references are not related to postnatal neurogenesis. For example, reference number 138 does not specifically discuss postnatal neurogenesis.

Author Response

This review covers postnatal neurogenesis, neurogenic zones, disorders with reduced neurogenesis, and bioactive compounds that have the potential to stimulate neurogenesis. Leveraging these compounds could offer novel strategies to improve cognition, address neurodegenerative conditions, and support brain health in general. The focus was on the beneficial effects of bioactive compounds such as Melatonin, resveratrol, EGCG, Quercetin, atranorin, Rg1, Saffron, Lavender, Sweet Basil, and Rosa damascena on brain health. Although the article is well-written and provides an in-depth account of various bioactive compounds, the following are important suggestions to improve the article:

A detailed account of bioactive compounds may be written as subsections 2.14.1, 2.14.2, and so on.

  • Thank you very much for this point. The changes are labeled in yellow.

Lines 67 to 117 give an account of neurogenesis in the DG. It would be great to add a brief account of neurogenesis in the olfactory bulbs and its roles (a brief of section 2.5).

  • We described the role of OB more deeply (see in yellow).

It would be beneficial to cite more original research and review articles in reputed journals rather than citing books so that readers could cross-validate the references. It is hard to verify book citations for readers.

  • Thank you very much for this point. In our manuscript, more than 165 references were cited. Eight from them are books or book chapters. For future work, we will pay more attention to cite original research.

Line 89: The process of postnatal neurogenesis, the period before the daughter cell becomes a mature neuron, lasts approximately 6-8 weeks [19]. This statement is not clear as it doesn’t mention if the experiments were done on humans or animals.

  • After reviewing more literature, we transformed the sentence as it was hard to set the exact period, because it is different (as you mentioned) in mice, rats, or humans, and some different authors mention different period. Thus, we decided to not specify the concrete maturation period. The new sentence is labeled in yellow.

Please add all of the following compounds (Melatonin, resveratrol, EGCG, Quercetin, atranorin, Rg1, Saffron, Lavender, Sweet Basil, and Rosa damascena) in the table; some of them are missing.

  • The table is created only for the compounds, not for the extracts from plants. The extracts are composed of many biologically active substances that have usually a mutual effect. This was the reason for not giving the plants into the table. We rewrote a bit this section of the manuscript and added more details about the plant extracts and bioactive compounds separately, so now it will be clearer to see that the table belongs to the compounds.

Some of the references are not related to postnatal neurogenesis. For example, reference number 138 does not specifically discuss postnatal neurogenesis.

  • Thank you. We studied the manuscript more carefully after your recommendation. The mentioned study (now the reference Nr. 155) already in the abstract part describes hippocampal neurogenesis, as Rg1 increased hippocampal cell proliferation.

Reviewer 2 Report

Comments and Suggestions for Authors

Authors presented the effects of neurotropic compounds on the postnatal neurogenesis.

1. Authors should include more details of the targets of each compounds in the table and also draw the involved/unified pathways in a figure.

2. It will be interesting to included monitoring biomarkers for each compounds to back up the effectiveness of each compounds in each mentioned animal models.

3. There are several more effective bioactive compounds in the postnatal neurogenesis, which should be included.

Author Response

Authors presented the effects of neurotropic compounds on the postnatal neurogenesis.

  1. Authors should include more details of the targets of each compounds in the table and also draw the involved/unified pathways in a figure.
  2. It will be interesting to included monitoring biomarkers for each compounds to back up the effectiveness of each compounds in each mentioned animal models.

- Thank you very much. We will discuss both points together. Many studies include the precise target of the compound studied, as well as biomarkers. However, many of them are focused on something different and hippocampal neurogenesis is studied only by mean of counting the amount of proliferatively active cells, such as the number of BrdU, Ki67, or DCX-positive cells. They are not focused on the background why it happens. That´s why it is really hard to find out a concrete target of the compound or biomarkers monitored. However, we added as much information as possible from all of the studies included in the manuscript and added the data into the table.

  1. There are several more effective bioactive compounds in the postnatal neurogenesis, which should be included.

- Thank you very much for this point. Plants are the source of various photochemicals; metabolites are used in medicinal and environmental sectors as well as being widely used in commercial and pharmaceutical products. Only from the group of flavonoids, more than 4000 different compounds have been identified from plant origin to date, as reviewed in Twaij et Hasan, 2022 (https://www.mdpi.com/2037-0164/13/1/3). It is hard for us to estimate which ones are more effective to be included in the manuscript. If there is a need to add some specific compound that you mean, please, let us know and we will be happy to add it. We added the information about the quantity of bioactive compounds into the manuscript (see in yellow).

Reviewer 3 Report

Comments and Suggestions for Authors

Review of the manuscript entitled: Bioactive compounds and their influence on postnatal neurogenesis. The manuscript touches on a very important topic but corrections need to be made.

The first section (Postnatal neurogenesis) is very long and much of the information seems unnecessary. Consider shortening this chapter. Moreover, the introduction (I think this is this section) should end with the aim of the work. e.g. "The aim of the present study was to ...".

Please check the entire manuscript carefully as it has many shortcomings e.g. lines 67, 75 and 104 “subgranular zone (SGZ)”. Why do you explain the abbreviation three times? what's more, it's similar with others. Please be precise, we only explain the abbreviation once.

References are missing in line 107

I believe that the content between lines 18 to 447 is not important in the context of the title (Bioactive compounds and their influence on postnatal neurogenesis). Shorten this content or change the title of the manuscript. The Authors should focus more on the substances which describes.

Table 1 is badly formatted, it should be clearer. It's hard to read now.

The conclusions should also include perspectives and information on potential applications.

Author Response

Review of the manuscript entitled: Bioactive compounds and their influence on postnatal neurogenesis. The manuscript touches on a very important topic but corrections need to be made.

The first section (Postnatal neurogenesis) is very long and much of the information seems unnecessary. Consider shortening this chapter. Moreover, the introduction (I think this is this section) should end with the aim of the work. e.g. "The aim of the present study was to ...".

  • Thank you. We added the aim of our review et the end of the chapter 1. See in yellow. We also shortened the chapter.

Please check the entire manuscript carefully as it has many shortcomings e.g. lines 67, 75 and 104 “subgranular zone (SGZ)”. Why do you explain the abbreviation three times? what's more, it's similar with others. Please be precise, we only explain the abbreviation once.

-Thank you for your point. We carefully checked all abbreviations, so now they have to be OK.

References are missing in line 107.

  • Thank you. We added the reference.

I believe that the content between lines 18 to 447 is not important in the context of the title (Bioactive compounds and their influence on postnatal neurogenesis). Shorten this content or change the title of the manuscript. The Authors should focus more on the substances which describes.

  • Postnatal neurogenesis and the regions, where it occurs, is very important when discussing its effects during various pathological conditions. That´s why we also described the diseases. We shortened the chapters (see in yellow).

Table 1 is badly formatted, it should be clearer. It's hard to read now.

  • Thank you. The redaction made this type of Table, we reformatted the table, it should now be more readable.

The conclusions should also include perspectives and information on potential applications.

  • We rewrote the conclusions of the manuscript focusing on perspective and potential applications.

Round 2

Reviewer 1 Report

Comments and Suggestions for Authors

The manuscript titled "Bioactive Compounds and Their Influence on Postnatal Neurogenesis" appears to be in excellent condition. The authors have implemented sufficient changes as per the suggestions.

Reviewer 3 Report

Comments and Suggestions for Authors

I accept the corrections